# Pressure-Induced Exciton Formation and Superconductivity in Platinum-Based Mineral Sperrylite

**DOI:** 10.3390/ma17143476

**Published:** 2024-07-13

**Authors:** Limin Wang, Rongwei Hu, Yash Anand, Shanta R. Saha, Jason R. Jeffries, Johnpierre Paglione

**Affiliations:** 1Maryland Quantum Materials Center, Department of Physics, University of Maryland, College Park, MD 20742, USA; 2Lawrence Livermore National Laboratory, 7000 East Avenue, Livermore, CA 94550, USA; 3Canadian Institute for Advanced Research, Toronto, ON M5G 1Z8, Canada

**Keywords:** superconductivity, exciton, pressure

## Abstract

We report a comprehensive study of Sperrylite (PtAs_2_), the main platinum source in natural minerals, as a function of applied pressures up to 150 GPa. While no structural phase transition is detected from pressure-dependent X-ray measurements, the unit cell volume shrinks monotonically with pressure following the third-order Birch–Murnaghan equation of state. The mildly semiconducting behavior found in pure synthesized crystals at ambient pressures becomes more insulating upon increasing the applied pressure before metalizing at higher pressures, giving way to the appearance of an abrupt decrease in resistance near 3 K at pressures above 92 GPa consistent with the onset of a superconducing phase. The pressure evolution of the calculated electronic band structure reveals the same physical trend as our transport measurements, with a non-monotonic evolution explained by a hole band that is pushed below the Fermi energy and an electron band that approaches it as a function of pressure, both reaching a touching point suggestive of an excitonic state. A Lifshitz transition of the electronic structure and an increase in the density of states may naturally explain the onset of superconductivity in this material.

## 1. Introduction

The enhancement of superconductivity as a function of applied pressures, known as the method to achieve the record-high transition temperature $T_{c}$ of 164 K at ∼31 GPa in the mercury-based cuprate superconductors [1], has recently come into strong focus due to the discoveries of near-room temperature superconductivity in hydride materials [2]. It has also long been a standard method of finding enhanced superconductivity in a wide range of compounds, including elements [3], heavy fermions [4], topological insulators [5], and oxides [6]. However, despite efforts to search for superconductivity in natural minerals [7], there are very few reports [8].

Sperrylite with formula PtAs_2_, named after Francis Louis Sperry, an American chemist in the late 1890s, forms in the cubic Pa-3 crystal structure shown in the inset of Figure 1, with Pt and As atoms occupying the 4a and 8c *Wyckoff* positions of the unit cell. Natural Sperrylite is tin-white, crystallized and is known to be metallic, with indistinct cleavage on 001 planes. Recent studies of PtAs_2_ and the related material PtSb_2_ have focused mainly on their thermoelectric properties. Attributed to the presence of a corrugated flat electronic band [9], PtSb_2_ reaches a maximum power factor of 43 µW/cmK^2^ at 400 K [10] with Ir partially substituted. A higher value of 65 µW/cmK^2^ at 440 K [11] was obtained by Rh substitution into PtAs_2_. Recently, the related pyrite material PtBi_2_, which was predicted to be a three-dimensional Dirac semimetal [12], displayed superconductivity under very large applied pressures [13] that appeared to alter the materials electronic structure to be nearly compensated. Overall, the Pt-based pyrite system hosts several intriguing properties and presents sensitivity to unit cell density that warrants further investigation. In particular, a change in the electronic structure as a function of lattice density can instill several rich phenomena that are important to consider. A change in the topology of the electron Fermi surface, known as a Lifshitz transition, occurs when the electronic band structure evolves through a topological change in its momentum-resolved geometric shape, and can give rise to very rich transport and thermodynamic phenomena [14].

Here, we have conducted high-pressure measurements on synthesized PtAs_2_ single crystals up to 150 GPa in order to investigate whether superconductivity can also be induced by an increase in the unit cell density, as well as to understand how electronic structure evolves. We find that the very small gap semiconducting behavior that occurs at ambient pressure is non-monotonically modified by applied pressures, giving way to a superconducting transition that emerges above 90 GPa and onsets as high as 3.5 K at the highest explored pressure of 150 GPa. We consider the evolution of the crystal structure with pressure and study the electronic structure to reveal a possible emergence of excitonic insulator behavior as well as a transformation to a new electronic structure that is supportive of superconductivity.

## 2. Materials and Methods

Naturally occurring Sperrylite mineral samples were obtained from the Department of Mineral Sciences at the Smithsonian Museum of Natural History as part of a broad search for superconductivity in natural minerals [7]. However, since no superconductivity was detected, we proceeded with a focus on lab-synthesized single-crystal samples of PtAs_2_ in order to characterize the material’s intrinsic properties and minimize impurity effects. Crystals were grown using a standard molten flux technique from lead flux with ratio PtAs_2_:Pb = 1:20. This mixture was heated to 1100 °C in an alumina crucible over 5 h, then cooled to 600 °C over 60 h, subsequently removing crystals from the melt by centrifuging the lead flux. Typical dimensions of obtained single crystals varied from several millimeters to over 1 cm, and samples were polished to ∼(2 × 1 × 0.1) mm^3^ for transport measurements.

Electrical transport and X-ray diffraction experiments under high pressures utilized diamond anvil cells (DACs) to generate pressures in excess of 100 GPa. The electrical transport DAC was constructed with a non-magnetic (BeCu) screw-driven cell body, and the high-pressure chamber comprised a 300 µm standard anvil, a non-magnetic MP35N gasket with a 90 µm hole, and a 260 µm, 8-probe designer anvil. The mismatched anvil sizes resulted in significant damage to the exterior of the culet of the designer anvil after the experiments; the CVD-grown diamond exhibited a ring-shape gouge approximately 300 µm in diameter, the same size as the culet of the opposing anvil in the experiment. Pressure in the chamber was measured via ruby fluorescence. Electrical transport data were acquired with a 4-probe technique using an AC resistance bridge to determine resistivity as a function of temperature using a commercial cryostat.

The DAC used for X-ray diffraction consisted of an asymmetric, steel piston–cylinder cell body combined with two opposed anvils with 250 µm diameter culets compressing a rhenium gasket pre-indented to approximately 35 µm in thickness. A 65 µm hole was drilled into the rhenium gasket using an electric discharge machine. Copper powder, a few ruby spheres, and the powdered PtAs_2_ were loaded into the pressure chamber before gas-loading the pressure chamber with neon as the pressure-transmitting medium. X-ray diffraction data were collected at the APS/HPCAT 16 BM-D beamline using a transmission geometry with a 29.2 keV beam aligned along the compression axis of the anvils. The X-ray beam was micro-focused to a spot size of 6 × 17 µm, and the detector was calibrated with a CeO_2_ standard. Pressure was increased using a gas membrane that was included in the DAC. Pressure was calibrated from the equation of state of copper [15]. Analysis of the X-ray diffraction data was carried out using Fit2D [16] and EXPGUI/GSAS-I [17].

Electronic structure calculations were obtained via first-principles density functional theory calculation using the WIEN2K [18] implementation of the full potential linearized augmented plane wave method within the PBE generalized gradient approximation. We used the lattice parameters from Ref. [19]. The *k*-point mesh was taken to be 14 × 14 × 14. To simulate the pressure effects on the electronic structure of PtAs_2_, the band structure and density of states calculations were then performed utilizing the same structure but adjusting the ambient pressure cubic lattice constant $a_{0}$ by factors of 0.9, 0.873, and 0.87 to mimic applied pressures of 123, 186 and 194 GPa, respectively, following the measured compression as described below.

## 3. Results and Discussions

Figure 1 presents powder X-ray diffraction data measured using the synthetic PtAs_2_ crystals at ambient pressure, with fitting performed using the previously reported pyrite structure [19]. Refinement of the diffraction spectrum yields a lattice constant of 5.9752(2) Å, which is consistent with the previous results [20]. Powder X-ray diffraction spectra of PtAs_2_ measured in the DAC were fit to extract unit cell dimensions as a function of pressure, plotted as volume (i.e., lattice parameter $a^{3}$) in Figure 2. PtAs_2_ does not undergo any structural phase transition within the measured pressure range, rather showing a continuous decrease in the unit cell volume as the pressure increases to 150 GPa. This evolution follows the Birch–Murnaghan isothermal equation of state (EOS): (1)$$P\left(V\right)=\frac{3B_{0}}{2}\left[{\left(\frac{V_{0}}{V}\right)}^{7/3}-{\left(\frac{V_{0}}{V}\right)}^{5/3}\right]\times \left[1+\frac{3}{4}\left(B_{0}^{\prime }-4\right)\left({\left(\frac{V_{0}}{V}\right)}^{2/3}-1\right)\right],$$
where *P* is the pressure, $V_{0}$ is the reference volume, *V* is the deformed volume, $B_{0}$ is the bulk modulus, and $B_{0}^{\prime }$ is the derivative of the bulk modulus with respect to pressure. Fitting the experimental data in Figure 2 to this model yields values $B_{0}=224\;\pm \;3$ GPa and $B_{0}^{\prime }=3.62\;\pm \;0.07$ for the bulk modulus and derivative term, respectively. The resultant fit is displayed in Figure 2 along with a comparison to the previously measured data [20]. The EOS provides an excellent fit, while the comparison to previous work shows a slight discrepancy which could possibly be due to the use of natural mineral specimens by Tschauner et al., which are likely not as pure as the crystals synthesized for this study. The previous study also claimed a better fit using the Vinet equation due to the small pressure derivative of the bulk modulus; however, the excellent fit to the Birch–Murnaghan EOS shown in Figure 2 suggests the value of $B_{0}$ obtained in the present study is more valid for pure PtAs_2_.

A study of the magnetic susceptibility for synthesized PtAs_2_, presented in Figure 3a, shows paramagnetic behavior on both warming and cooling curves, with no evidence of magnetism. The lack of temperature dependence is similar to previous reports for PtAs_2_ as well as RhAs_2_ and IrAs_2_ [21], although the historical study reports a diamagnetic response. Figure 3b presents the electrical resistivity of PtAs_2_ at ambient pressure, which is also lacking any significant features. At high temperatures, the transport behavior is that of a semimetal where resistivity is nearly flat, exhibiting a slight decrease on cooling down to a broad minimum centered near 200 K, followed by an increase with moderately activated behavior. Fitting to a standard Arrhenius activated behavior using $\rho \left(T\right)=\rho \left(0\right)e^{\Delta /k_{B}T}$ through the range 50–200 K as illustrated in the inset, we obtain a rather small value for the thermally activated energy gap Δ = 3.96 meV. Below 50 K, ρ(T) deviates from activated behavior, which indicates that there may be another conduction channel or scattering mechanism at low temperatures that requires further investigation.

Figure 4 presents the resistance temperature dependence of a PtAs_2_ crystal mounted in the DAC cell and pressurized up to 150 GPa. The R(T) data exhibit a non-monotonic evolution as a function of applied pressure, with the lowest applied pressure data (4.2 GPa) exhibiting semiconducting behavior similar to that of ambient pressure that evolves to a more insulating behavior at mid-range pressures before becoming more metallic at higher pressures. As shown in the inset in Figure 4, an abrupt drop in resistivity below about 4 K emerges at pressures above 77 GPa, which we attribute to the emergence of superconductivity. Due to the extremely high pressures of this phase, it is difficult to perform other experiments to confirm the superconducting state. However, the increasing drop that is evident with increasing pressure is consistent with such a state. Given the appearance of superconductivity near 2 K in the related pyrite material PtBi_2_ near ∼10 GPa [13], it is not surprising that a similar onset occurs in PtAs_2_, albeit at much higher pressures. However, we note that the purported electronic structures are quite different as discussed below, so further work is required to understand the relationship between the two materials.

The evolution of resistivity with applied pressure is quantified in Figure 5. As shown in panel (a), the non-monotonic evolution is evident when comparing the evolution of resistance values at 2 K and 300 K, which exhibit different dependence as a function of pressure. Normalizing the absolute changes by plotting their ratio in panel (b) reveals a striking peak near ∼20 GPa. The temperature dependence of resistivity qualitatively changes with pressure evolution and exhibits deviations from simple semiconductor-like activated behavior at low temperatures, suggesting there is a more complex behavior that is not captured by a simple Arrhenius function. To quantify this evolution, we apply an empirical power law fit of the form
(2)$$R\left(T\right)=\frac{1}{G_{0}+aT^{n}},$$
to track the pressure evolution, where $G_{0}$ is the residual zero-temperature conductance, *a* is a scaling coefficient for the temperature (*T*) dependence, and *n* is the power law exponent. In Figure 5c, we see the pressure effect on the conductance model described in Equation (Equation 1), which shows the evolution from a simpler $T^{-1}$ power law at low pressures to a stronger behavior with a peak in the power law exponent *n* at a similar pressure to the resistance ratio plot in panel (b). While the power law fits deviate from the measured data, this model is meant to capture a general phenomenological trend rather than fit a specific model. Interestingly, the power law increases and plateaus close to n=1.5, which is the power law first calculated by Wilson in 1931 for the mobility of a system due to acoustic phonon scattering [22]. However, this is a very simplified behavior derived using a single parabolic band and a dispersionless phonon frequency, and more realistic calculations point to power laws ranging between linear and cubic [23]. Furthermore, as shown in Figure 5c, *n* subsequently decreases to a much smaller value of ∼0.5 at higher pressures. Since the phonon spectrum is not expected to significantly change with pressure, and this power law behavior is observed down to temperatures much below the Debye scale where thermal activation is thought to dominate, it suggests a more complex behavior. Below, we discuss possible sources of this non-monotonic behavior tied to the evolution of the electronic band structure.

In Figure 6, we present the band structure and density of states for PtAs_2_, which appears to be consistent with database calculations [24] that indicate a trivial topological structure, unlike that of PtBi_2_ [12] and likely due to the gapped nature of the band structure and lighter pnictogen anion in PtAs_2_. We study the evolution of the PtAs_2_ band structure as a function of cubic unit cell densities calculated for lattice constants of $a_{0}$, $0.9a_{0}$, $0.873a_{0}$ and $0.87a_{0}$ with respect to the ambient pressure lattice constant $a_{0}$. These densities correspond to the estimated applied pressures of 0, 123, 186 and 194 GPa as calculated using the unit cell volume dependence measured by X-ray diffraction shown in Figure 2. As is evident, there is a hole band centered between the Γ and M Brillouin zone points that remains very close to the chemical potential at all densities, an electron pocket at R that rapidly moves higher in energy (i.e., further from the chemical potential), while another electron pocket at Γ gradually drops toward the chemical potential. Together, this evolution of the band structure is broadly consistent with the measured electrical transport behavior, including the non-monotonic evolution of its temperature dependence as discussed below.

As shown in Figure 6a, the ambient pressure hole band (blue) centered between Γ and M points just barely cuts across the Fermi level, reflecting a non-metallic semimetal behavior consistent with our experimental observation. By applying pressure, this band is very subtly pushed down below the chemical potential, opening an indirect semiconducting gap between this band and the electron band centered at the Γ point (red). This change in the hole band is hard to discern when comparing to the first high-pressure panel for $a=0.9a_{0}$ (123 GPa) but is clear when we plot the position of the band edges versus pressure with higher point density in Figure 7. As shown, the calculated hold band edge crosses the chemical potential at approximately 20 GPa effective pressure, which is precisely where we observe a non-monotonic change in the evolution of transport features (see Figure 5b). Thus, this semimetal–semiconductor crossover in the low-pressure regime provides a good explanation of our transport data in the same pressure range.

With increasing pressure, e.g., from $a=0.9a_{0}$ (123 GPa) to $0.837a_{0}$ (186 GPa), the indirect gap narrows, and eventually the conduction and valence bands touch the Fermi level at the same time, very close to $a=0.873a_{0}$ as shown in Figure 6c. This approach results in a situation favoring the formation of an exciton insulator state [25], where the formation of electron–hole pairs occurs with a binding energy typically within tens of meV. In PtAs_2_, the pressure-induced shrinking of the energy gap toward the touching point at $a=0.873a_{0}$ provides ripe conditions for a thermodynamically stable excitonic state, which should, in principle, initiate the onset once the gap energy falls below the excitonic binding energy. Without further experiments, it is unclear what energy range is required to achieve this condition, but the experimental transport data shown in Figure 5b clearly mark a trend toward smaller and smaller activation energies above the ∼20 GPa resistance maximum, with a plateau reached above approximately 50 GPa. More interestingly, the calculated pressure of the band touching point is very near the pressure where we observed a sudden drop in resistance at low temperatures, which we attribute in the discussion above to an onset of superconductivity. It is certainly tempting to consider whether there is a correlation between these two events, and whether a more exotic type of pairing mechanism may be at play. Excitonic pairing mechanisms have long been proposed as a possible route to superconductivity, and mostly focus on a proximitized excitonic medium [26]. However, more recent proposals have suggested an intrinsic pairing mechanism is possible that involves a spin-triplet model that avoids strong Coulomb repulsion issues [27]. What relation the band structure, superconductivity, and exciton formation have requires further study to elucidate.

In any case, the interesting evolution of the band structure at least entails several notable topological changes. These so-called Lifshitz transitions are known to give rise to interesting changes in the electronic and physical properties as shown in the iron [28] and nickel pnictides, for example, in the electronic nematic system BaNi_2_As_2_ [29], where a Lifshitz transition appears to be associated with phase transitions involving superconductivity.

As shown in Figure 6 and Figure 7, there are at least two notable Lifshitz transitions that occur as PtAs_2_, which we attributed to measured changes in transport properties. This is also observed in the evolution of the electronic density of states (DOS) at the Fermi level (see Figure 6), which first decreases and then increases with increasing pressure. Therefore, both the evolution of the band structure and DOS reveal the same physical trend as shown in Figure 5. Furthermore, the components of the DOS plotted in Figure 6 exhibit an interesting evolution that entails a valence valence band mainly composed of As 3*p* orbitals and a conduction band made up of Pt 5*d* orbitals at ambient pressure: upon increasing pressure, the Pt-5*d* bands are pushed away from the Fermi level, and eventually, the bands around the Fermi level are dominated by As-3*p* orbitals, which no longer have a covalent character but appear to be metallic. At the highest pressures, the higher DOS at the Fermi level could in principle also play a role in stabilizing superconductivity, but to understand how exciton formation also plays a role will require a microscopic model and further study.

## 4. Conclusions

In conclusion, we present structural and electrical transport properties of the platinum-based mineral Sperrylite (PtAs_2_) as a function of applied pressures up to 150 GPa, and compare our results with electronic structure calculations as a function of the cubic unit cell lattice density to help elucidate an unusual non-monotonic evolution of transport from semiconducting to metallic at high pressures. As opposed to the monotonic evolution of the unit cell density with pressure, which is well described by a third-order Birch–Murnaghan equation of state, the calculated evolution of electronic bands—from semimetallic to insulating to metallic with increasing pressure—well explains the evolution of electrical transport, and suggests an excitonic instability may arise near the high-pressure band-touching point. At the highest pressures, the observation of a sudden drop in resistance at 3 K suggests the onset of superconductivity very close to the calculated band-touching point, suggesting a strong interplay between the excitonic band structure evolution and appearance of superconductivity in this compound.

## Figures and Tables

**Figure 1 materials-17-03476-f001:**
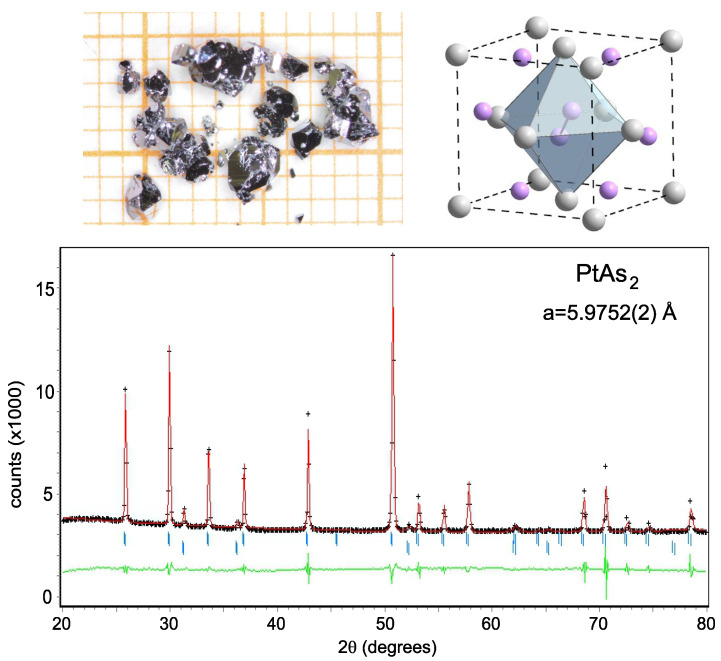
X-ray diffraction data from powdered PtAs_2_ single crystals, with a small amount of lead identified as originating from the growth flux. Black crosses, red lines, and green lines represent the observed data, fitting curve and the difference between the observed and fit (residuals), respectively. The insert shows the unit cell of PtAs_2_, with a Pa-3 cubic crystal structure (grey and pink spheres represent Pt and As atoms, respectively.

**Figure 2 materials-17-03476-f002:**
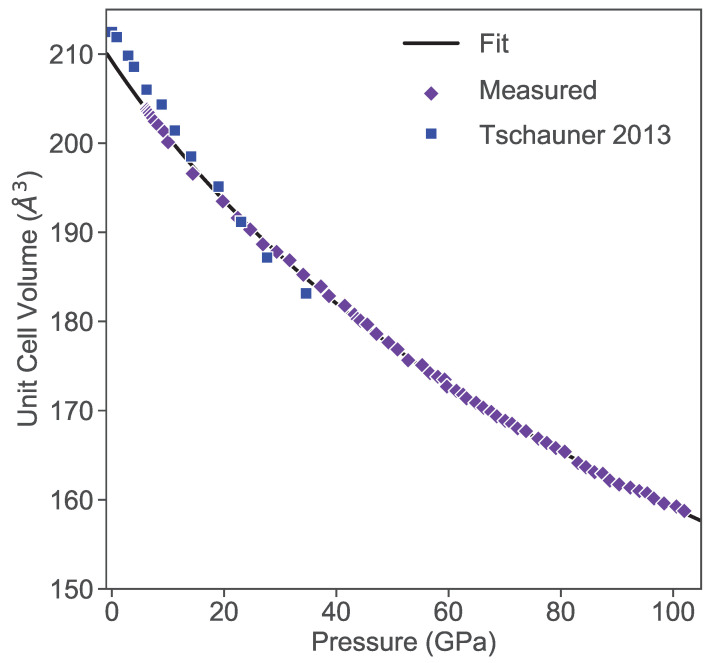
Cubic unit cell volume of PtAs_2_ obtained by X-ray diffraction under applied pressures. Data from this study (purple diamonds) are fit (solid line) to the Birch–Murnaghan isothermal equation of state as described in the text, and compared to data measured in a previous study (blue squares) by Tschauner et al. [20].

**Figure 3 materials-17-03476-f003:**
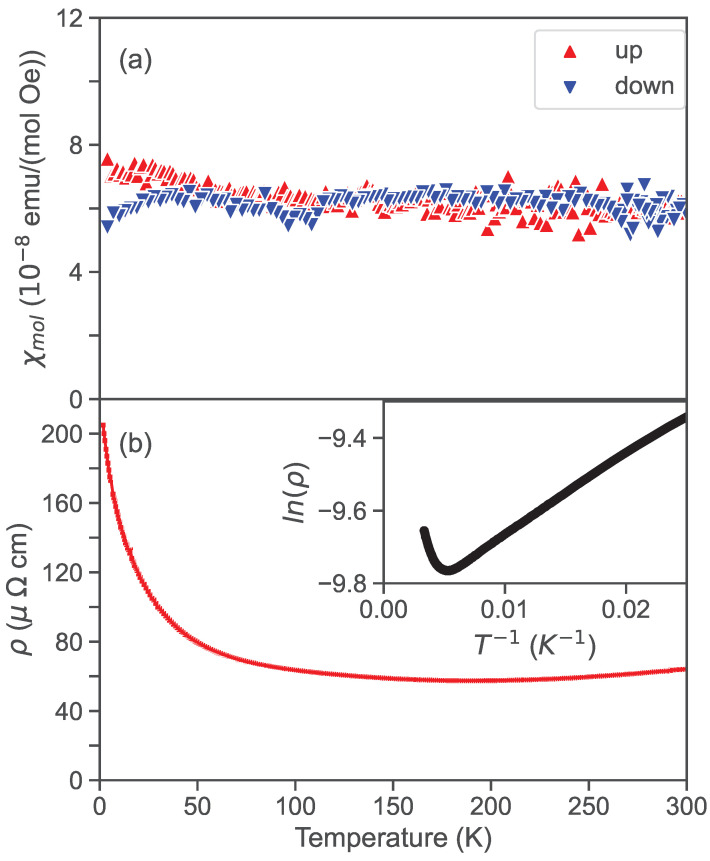
(**a**) Temperature-dependent magnetic susceptibility measured at 1 T of a lab-synthesized PtAs_2_ single crystal. (**b**) Electrical resistivity temperature dependence of lab-synthesized PtAs_2_ single crystal at ambient pressure. Inset shows Arrhenius plot of data, indicating semiconducting behavior at low temperatures.

**Figure 4 materials-17-03476-f004:**
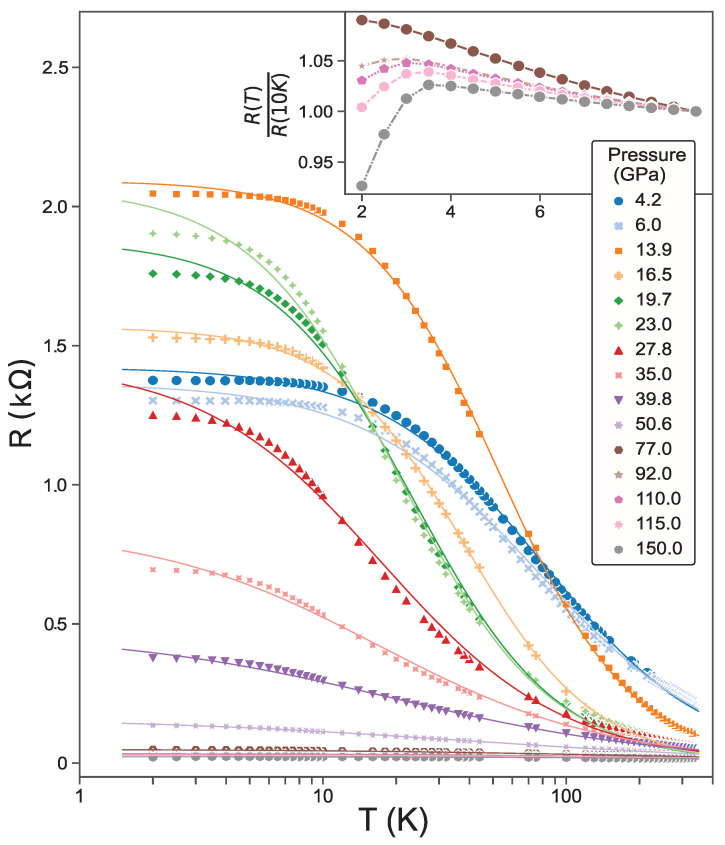
Resistance of single crystal PtAs_2_ as a function of applied pressures in the range 4.2–150 GPa. Data points represent experimentally measured data, and solid lines are the results of the phenomenological power law fit to Equation (Equation 2). The inset shows a zoom of the low-temperature resistance at pressures (top to bottom) of 77, 92, 110, 115, and 150 GPa, respectively, highlighting the drop in resistance due to the onset of a superconducting state.

**Figure 5 materials-17-03476-f005:**
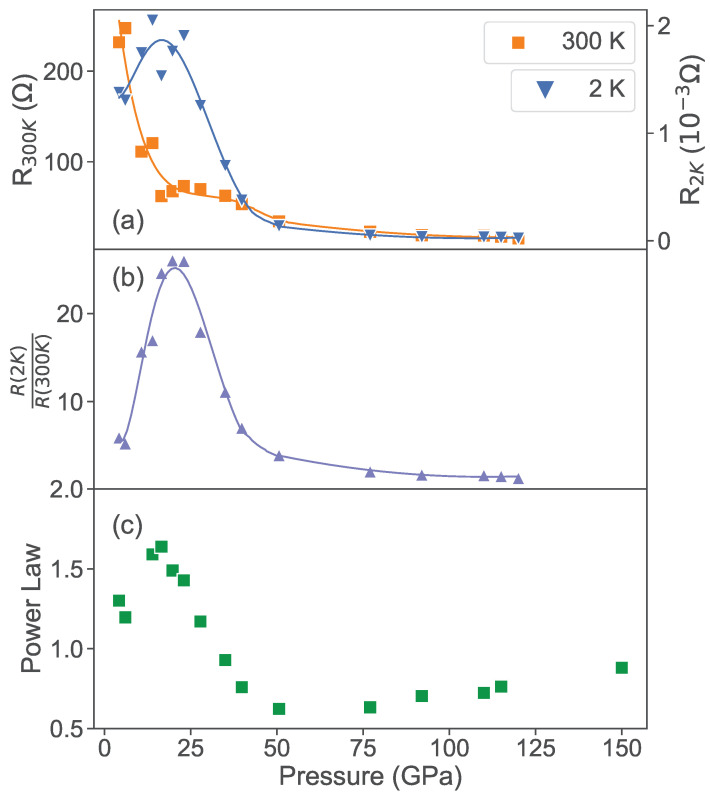
Evolution of PtAs_2_ sample resistance features as a function of applied pressure, shown (**a**) at room temperature (orange squares) and 2 K (blue triangles), and (**b**) as a ratio (purple triangles). (**c**) Evolution of the temperature power law exponent *n* from fits to conductance model $R\left(T\right)=1/\left(G_{0}+aT^{n}\right)$ as a function of pressure as explained in the text.

**Figure 6 materials-17-03476-f006:**
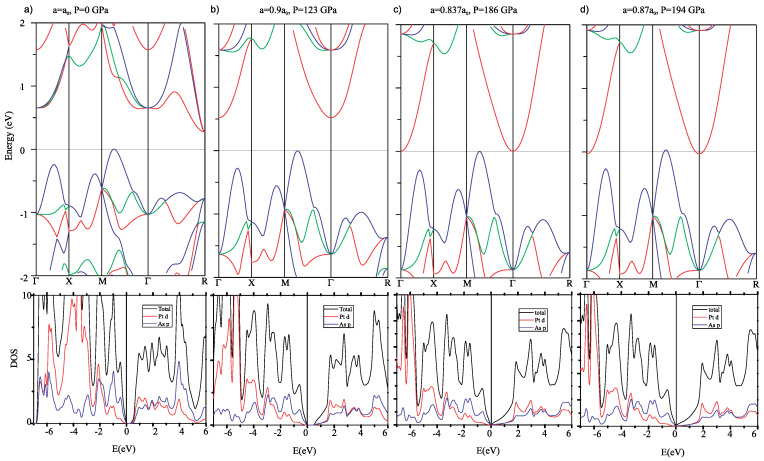
Low-energy electronic band structure (top panels) and electronic density of states (bottom panels) of PtAs_2_ as a function of unit cell density, using lattice parameters of $a=1.0a_{0}$ (**a**), $a=0.90a_{0}$ (**b**), $a=0.873a_{0}$ (**c**), $a=0.87a_{0}$ (**d**), where $a_{0}$ is the lattice constant for the ambient pressure unit cell. The corresponding value of pressure P for each calculation is estimated based on Equation (Equation 1). Colors in panels (**a**–**d**) represent distinct band contributions.

**Figure 7 materials-17-03476-f007:**
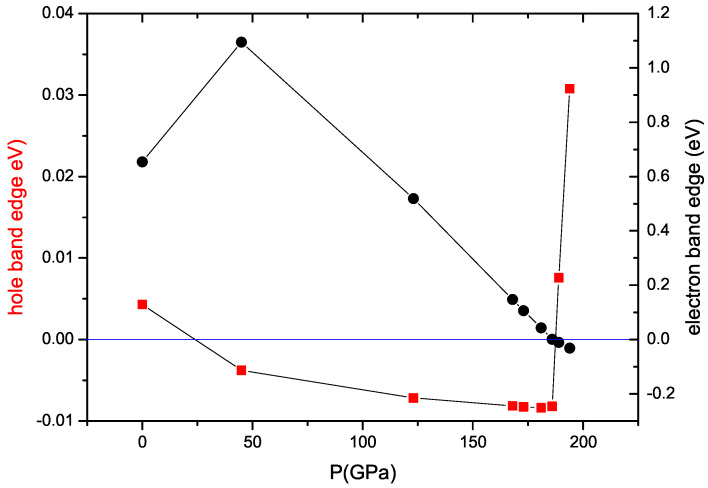
Position of electron (circles) and hole (squares) band edges with respect to the Fermi energy (E = 0) as a function of calculated pressure from DFT band structure calculations.

## Data Availability

The raw data supporting the conclusions of this article will be made available by the authors on request.

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
