# Peer review of "Pressure-Induced Exciton Formation and Superconductivity in Platinum-Based Mineral Sperrylite"

_materials, 2024, doi:10.3390/ma17143476_

Round 1

Reviewer 1 Report

Comments and Suggestions for Authors

The article´s strength is finding possibilities to create superconductivity materials from natural minerals under very high pressure. It is tightly written in two columns and seven pages.

1.       No one references numbers inside the text and in the reference chapter, and there are no figure numbers inside the text. It needs to be set up all.

2.       It needs to explain in the introduction what are the properties of the material  Sperrylite

3.       The signing of  Fig. 2 mentions dots on the surface of the figures. I have not seen these.

4.       What are the structure properties of the Pa-3 crystal structure present in Fig. 1?

5.       Figs is not present in signing capital letters but by Fig. x?

6.       What happened physically inside the structure of  Sperrylite? Iti s not described.

Reviewer 2 Report

Comments and Suggestions for Authors

This experimental paper focuses on the study of the effect of pressure on the properties of Sperrylite and the occurrence of superconductivity in it. Frankly speaking, although I see some interesting results at the beginning of the paper, the further part devoted to superconductivity seems very doubtful. That is why my decision is to reject this publication. 

Below are my main comments:

1) the emergence of superconductivity is confirmed not only by resistive measurements but also by magnetic susceptibility measurements. In both cases I see neither a clear drop of resistivity to zero nor the occurrence of the known -1/(4*pi) for the magnetic susceptibility.

2) The mention of the Lifshitz transition is completely unjustified. I recommend the authors to read the review devoted to topological Lifshitz transitions in superconductors https://pubs.aip.org/aip/ltp/article-abstract/47/8/672/252650/Concise-guide-for-electronic-topological?redirectedFrom=fulltext

3) Technical note: reading of the paper is complicated by an unsuccessful compilation in Latex, because instead of references there are only question marks. 

Reviewer 3 Report

Comments and Suggestions for Authors

The authors studied the electrical characteristics of PtAs2 under various pressures. The research about various characteristics under high pressure. The authors discussed the results by performing the fitting analysis. This discussion is helpful for the readers studying the superconductor. However, I am concerned about the introduction and the results. If the authors appropriately revise the manuscript, this study will meet the criteria for the publication in materials.

Comment list

Comment 1: In Figure 5a, the pressure at which R reaches the maximum value becomes higher when decreasing temperature from 300 to 2 K. Why does this happen?

Comment 2: Please describe the physical meaning of power law. For example, in the semiconductor, the power law of 1.5 indicates phonon scattering.

Comment 3: The introduction is too specific. The present manuscript has a lack of the preceding studies about phase transition. The superconducting transition is one of the phase transitions. The authors should mention the background of phase transition for the readers to easily understand the motivation of this study. For example, there have been a lot of famous studies about structural phase transition: thermal switch (Nano Lett. 22, 6105 (2022).), phase change memory (ACS Appl. Mater. Interfaces 11, 5336 (2019).), etc. If the authors improve the introduction, the revised manuscript will attract the interest of a lot of readers.

Comment 4: The numbers of figure and reference are converted to “?”. Please revise it.

Round 2

Reviewer 1 Report

Comments and Suggestions for Authors

Artice is OK.

Reviewer 2 Report

Comments and Suggestions for Authors

I am grateful to the authors for significantly improving the manuscript and clarifying all my comments

Reviewer 3 Report

Comments and Suggestions for Authors

Everything was cleared.